# Point-of-care biomarkers for prediction of kidney function trajectory among sugarcane cutters: a comparative test accuracy study

Erik Hansson [1,2] Catharina Wesseling, [1,3] David Wegman, [1,4] Ulf Ekström, [1,5] Denis Chavarria, [6] Jason Glaser, [1] Kristina Jakobsson [1,2,7]

For numbered affiliations see end of article.

**Correspondence to**
Dr Erik Hansson;
erik.hansson@amm.gu.se

## ABSTRACT

**Objectives** Heat-stressed Mesoamerican workers, such as sugarcane cutters, suffer from high rates of chronic kidney disease of non-traditional origin (CKDnt). We aimed to identify easily available early markers of rapid kidney function decline in a population at high risk of CKDnt.

**Design** The accuracy of different biomarkers measured during harvest for prediction of cross-harvest kidney function decline were assessed in an exploratory study group, and the performance of the most promising biomarker was then assessed in an independent confirmation group.

**Setting** Male sugarcane cutters in El Salvador and Nicaragua.

**Participants** 39 male Salvadoran sugarcane cutters sampled fortnightly at ≤9 occasions before and after work shift during harvest. 371 male Nicaraguan sugarcane cutters were sampled as part of routine monitoring during two harvests. Cutters worked at high physical intensity at wet-bulb globe temperatures mostly above 29°C for 6–8 hours per day 6 days a week during the 5–6 months harvest season.

**Primary outcomes** Change in estimated glomerular filtration rate (CKD Epidemiology Collaboration) across the harvest season ($\Delta eGFR_{cross-harvest}$).

**Results** Dipstick leukocyturia after work shift in the El Salvadoran group was the most promising marker, explaining >25% of $\Delta eGFR_{cross-harvest}$ variance at 8/9 occasions during harvest. Leukocyturia was associated with experiencing fever, little or dark urine, cramps, headache, dizziness and abdominal pain in the preceding 2-week period. Decreasing blood haemoglobin (Hb) and eGFR during harvest were also predictive of $\Delta eGFR_{cross-harvest}$. In the Nicaraguan confirmation dataset, those having ≥++ leukocyturia at any sampling during harvest had a 13 mL/min/1.73 m$^2$ (95% CI 10 to 16 mL/min/1.73 m$^2$) worse $\Delta eGFR_{cross-harvest}$ than those without recorded leukocyturia.

**Conclusion** Leukocyturia and Hb, both measurable with point-of-care methods, may be early indicators for kidney injury and risk for eGFR decline among heat-stressed male workers, thereby facilitating individual-level prevention and research aiming to understand the causes of CKDnt.

## STRENGTHS AND LIMITATIONS OF THIS STUDY

⇒ The accuracy of a wide range of markers available as point-of-care analyses for early prediction of change in kidney function was explored in a population at high risk of chronic kidney disease of non-traditional origin.

⇒ The trajectory of kidney function change was observed longitudinally.

⇒ The main finding was replicated in an independent dataset collected routinely in an occupational setting.

⇒ The long-term effect on kidney health resulting from early action on kidney injury markers is not studied.

## INTRODUCTION

An epidemic of chronic kidney disease of non-traditional origin (CKDnt) has caused tens of thousands of premature deaths in Mesoamerica over the past decades.[1] Although the exact cause is unknown, research supports that physically demanding work in heat is a key driver of disease initiation and/or progression.[2]

High rates of abrupt increases in kidney injury markers, such as serum creatinine (SCr), coinciding with systemic inflammatory activity have been seen in working populations at high risk of CKDnt.[3 4] While this condition sometimes requires hospitalisation,[3] it may also be compatible with continued physical work, with active and apparently well sugarcane workers found to have raised SCr and C reactive protein (CRP) over a harvest season.[4] Monitoring workers for evidence of asymptomatic kidney injuries may be an effective way of preventing CKDnt as it could allow for early detection, risk factor elimination and treatment. Furthermore, monitoring workers for early kidney injury could be useful for epidemiological studies to understand CKDnt aetiology, as assessment of exposures is easier the shorter the recall. CKDnt has

been reported primarily in resource-constrained settings, meaning low costs and high logistic feasibility are key for any such screening procedures. Furthermore, minimally invasive procedures are desirable as this likely increases acceptance of repeat testing.

Markers of kidney function, such as creatinine, could give an indication of CKDnt prognosis.[5][6] The probable link between inflammation and initiation and progression of CKDnt[3][4] indicates that markers of inflammation such as leukocyturia, white blood cell count (WBC) and CRP elevation may function as early markers of CKDnt risk. There are previous reports of associations between leukocyturia measured using dipsticks and cross-shift SCr elevation in sugarcane workers,[7] acute kidney injury (AKI) among hospitalised sugarcane workers[3] and lower estimated glomerular filtration rate (eGFR) at end-harvest in a cross-sectional analysis.[8]

Anaemia or incident anaemia is a common finding in sugarcane workers with SCr elevation[3][4][9] and it indicates worse prognosis following AKI among sugarcane workers,[9] suggesting that a decline in haemoglobin (Hb) level may be useful for identifying workers with a high risk of renal disease progression.[9] The use of Hb as a marker of kidney disease risk is further supported by observations of declining erythropoietin levels co-occurring with declining Hb among sugarcane workers, possibly indicating that reduced renal erythropoietin synthesis secondary to tubulointerstitial injury could be a common cause of declining Hb in this setting.[10–12]

This study aims to explore whether some easily available and relatively cheap analyses (urine dipstick, creatinine and CRP in serum, Hb and WBC in blood) may be useful for identification of workers at risk of developing reduced GFR during a prolonged heat stress exposure. We do this by first analysing a dataset from Salvadoran workers and then confirming the findings with a second dataset from Nicaraguan workers performing the same work tasks. As a post hoc analysis, we explored the association between leukocyturia and a range of symptoms.

## METHODS
### Patient and public involvement
Patients and/or the public were not involved in the design, or conduct, or reporting, or dissemination plans of this research.

### Outcome
The cross-harvest change in preshift eGFR (ie, $\Delta eGFR_{cross-harvest} = eGFR_{end-harvest} - eGFR_{preharvest}$) was used as outcome variable. eGFR was estimated using the Chronic Kidney Disease Epidemiology Collaboration (CKD-EPI) equation.[13]

### Exploratory study group: the Worker Health and Efficiency Project
The Worker Health and Efficiency (WE) Project was a rest-shade-hydration intervention pilot study conducted at the Ingenio El Angel in El Salvador during the 2014–2016 sugarcane harvests.[14][15] The working conditions, including the heat stress intervention, have been previously reported, as well as descriptive biochemical results (eg, SCr, creatinine phosphokinase (CPK)) from before and after work shifts.[14][15] In the first harvest (2014–2015), the inland group of sugarcane cutters (n=49) provided urine samples that were analysed by urine dipsticks preshift and postshift every 2 weeks (up to nine occasions during harvest per worker). One female worker was excluded as biological differences (eg, menstruation and urinary tract infection (UTI) susceptibility) were expected to influence biomarker predictive performance and sex-stratified analyses were impossible due to too few women in the study. Results thus apply to men. Preshift and postshift blood samples (for CRP, WBC, SCr and Hb) were taken before, twice during and at the end of harvest (table 1). This dataset was used to explore markers potentially useful as early kidney disease indicators. These workers did not undergo any screening before hiring or evaluation by mill staff during harvest.

### Urine dipstick
Bayer Multistix 10 SG were read by a Siemens CLINITEK Status+ Analyzer.

Urine dipstick urine-specific gravity (USG) (intervals of 5 from 1005 to 1030), and haematuria (0, +, ++, +++), leukocyturia (0, +, ++) and proteinuria (0, trace, +, ++, +++) were entered as linear independent variables in separate regression models of $\Delta eGFR_{cross-harvest}$ at each of the nine sampling occasions during harvest and time of sampling (preshift and postshift). For each model, the proportion of the variance in $\Delta eGFR_{cross-harvest}$ explained by the dipstick parameter was calculated.

### Blood/Serum biomarkers
Blood samples were obtained by venipuncture. Blood Hb and WBC were measured at Laboratorio CECIAM Escalón, San Salvador. CRP and SCr were measured using a Cobas 701 instrument (Roche Diagnostics, Basel, Switzerland) at Lund University Hospital, Sweden. eGFR was estimated using SCr and the CKD-EPI equation.[13]

For Hb and eGFR, the difference between the morning preharvest (baseline) levels and those measured at the two preshift and postshift occasions during harvest ($\Delta Hb_{within-harvest}$, $\Delta eGFR_{within-harvest}$) were used as four independent explanatory variables for $\Delta eGFR_{cross-harvest}$, while the absolute values were used for the four during-harvest measures of CRP (log-transformed) and WBC. It was considered that changes in eGFR and Hb from preharvest levels are likely more relevant for identifying ongoing pathological process than isolated absolute values, whereas absolute levels of CRP and WBC were considered interpretable in isolation.

Each of the blood/serum biomarkers ($\Delta Hb_{within-harvest}$, $\Delta eGFR_{within-harvest}$, WBC and CRP) collected preshift and postshift on each of the two measurement days during the harvest (table 1) were entered separately as independent

**Table 1** Overview of data available from El Salvador and Nicaragua cohort studies[14–17]

| Exploratory dataset, El Salvador, WE Project | | | | | | | | | | |
|---|---|---|---|---|---|---|---|---|---|---|
| Preharvest | Week | | | | | | | | | End-harvest |
| | 2 | 4 | 6 | 8 | 10 | 12 | 14 | 16 | 18 | |
| Questionnaire, 2-week recall | | | | | | | | | | |
| Symptoms | X | X | X | X | X | X | X | X | X | X |
| Biomarker, preshift and postshift | | | | | | | | | | |
| Dipstick: SpG, RBCs, WBCs, protein | X | X | X | X | X | X | X | X | X | X |
| eGFR | | | X | | | | X | | | X |
| CRP | | | X | | | | X | | | X |
| WBC | | | X | | | | X | | | X |
| Hb | | | X | | | | X | | | X |

| Confirmation dataset, Nicaragua, Adelante | | | | |
|---|---|---|---|---|
| Preharvest | Cross-harvest | | | End-harvest |
| | Routine random monitoring of workers | Mid-harvest, ~week 10 | Routine random monitoring of workers | |
| eGFR | Dipsticks preshift and postshift as part of hydration monitoring (specific gravity). Leukocyturia is recorded | Creatinine screening aiming to identify workers with kidney injury for rest or ending or changing employment | Dipsticks preshift and postshift as part of hydration monitoring (specific gravity). Leukocyturia is recorded | eGFR |

RBCs=haematuria, WBCs=leukocyturia.
CRP, C reactive protein; eGFR, estimated glomerular filtration rate; Hb, haemoglobin; RBC, red blood cell; SpG, specific gravity; WBC, white blood cell; WE, Worker Health and Efficiency.

variables in linear regression models with $\Delta eGFR_{cross-harvest}$ as dependent variable. $R^2$ values for each marker and time of sampling were calculated as estimates of the predictive ability of each of the markers taken at specific time points in relation to work. Workers who did not provide samples at a visit were not included in the regression model for that specific visit.

## Confirmatory study group: the Adelante study

The finding that leukocyturia might be a good predictor of $\Delta eGFR_{cross-harvest}$ in the Salvadoran workers was studied further using data collected 2017–2019 via routine monitoring of all sugarcane cutters participating in the Adelante Intervention Study[16 17] at Ingenio San Antonio, Chichigalpa, Nicaragua (table 1).

In contrast to the Salvadoran sugar mill, the Nicaraguan sugar mill had a pre-employment screening, excluding workers with high preharvest SCr (>1.3 mg/dL for men, >1.0 for women), anaemia, diabetes, hypertension or elevated uric acid levels. All hired male cutters with preharvest and end-harvest serum samples in the first 2 years of the Adelante Study who had measurements of dipstick leukocyturia were included (n=451 worker-harvests). Again, women were excluded for the same reason the one female Salvadoran subject was—because their small number (14% available worker harvests) would result in inadequate statistical power.

Routine monitoring in the field was standard practice for health promoters trained and employed by the occupational health (OH) unit at the mill and consisted of monitoring a portion of workers each day for indicators of liquid intake, including urine dipsticks. Urine dipsticks (Combur 10) were machine-read and the data entered in an electronic system. Workers were selected according to a rolling schedule with the intention to include all workers at least once during the harvest. Those considered in the judgement of the health promoters to be in need of increased attention were selected for more frequent monitoring.

The routine in the mill was to supplement the health promoter urine monitoring by measuring SCr among all the workers at mid-harvest. Workers with SCr above employment thresholds were put on sick leave but could return to work if SCr decreased below threshold within a few weeks. If not, their employment may have been ended or they may have been transferred to another job group with lower workload, thus missing end-harvest sampling.[16 17] Workers were also continuously observed and assessed by health promoters who might refer seemingly unwell workers to a hospital at the mill, where SCr is measured when indicated,[3] something which might lead to ending employment.

All workers participating in the Adelante study had serum samples collected by OH staff before and at the end of harvest as part of the research project. These samples were analysed for SCr at Lund University Hospital, Sweden. Calculation of $\Delta eGFR_{cross-harvest}$ was based on these results only. Workers who did not attend end-harvest follow-up serum sampling were followed up

by answering a questionnaire on the reason they did not participate.

As in previous publications from the Adelante cohort,[16 17] we defined incident kidney injury ($IKI_{all}$) during harvest as a ≥0.3 mg/dL SCr increase among workers finishing harvest or by self-reported kidney injury during harvest among workers not finishing the harvest. As almost all dropouts were reached, this outcome suffers from minimal bias due to loss-to-follow-up and associated healthy worker effects. The proportion of workers with recorded leukocyturia during the harvest by $IKI_{all}$ status was calculated.

Leukocyturia +++ was recoded to ++ as this was the maximum in El Salvador and +++ was infrequent. The maximum dipstick postshift leukocyturia recorded during harvest for each worker was used as a continuous variable in a mixed linear regression model with cross-harvest eGFR as outcome variable. Random effects were included for worker as some (n=80) were included in both harvest seasons.

The vast majority of workers perform less physically strenuous work between harvests. In order to assess whether leukocyturia-associated eGFR loss is rapidly reversible on cessation of heavy harvest work, we calculated eGFR trajectories from preharvest year 1 to preharvest in year 2 and compared this by maximum postshift leukocyturia status during harvest 1 using linear regression.

### Was leukocyturia related to worker symptoms?

We also wanted to explore whether leukocyturia was associated with any symptoms, which could be additional useful information when designing screening programmes. The Salvadoran workers were asked how many of the 14 preceding days they had experienced any of 21 symptoms at the fortnightly urine samplings. This analysis included all male cutters in the job group, that is, also those not finishing harvest and also workers with postshift dipsticks taken at end-harvest, dipsticks which were not included in the above analysis of $\Delta eGFR_{cross-harvest}$ prediction. Postshift leukocyturia was modelled as an ordinal dependent variable and the days of experiencing symptoms both as a continuous variable, and dichotomised at NO versus ANY day using mixed effects ordered logistic regression using the Stata meologit command.

## RESULTS
### Exploratory analyses

Thirty-nine out of 49 male workers had baseline and end-harvest serum samples and were included. The mean number of urine dipsticks taken per worker during harvest was 7.1 (range 2–9) preshift and end-shift pairs. Seventy-seven per cent of workers had two paired preshift and end-shift blood samples taken during harvest, while 23% participated in blood sampling only once during harvest.

Leukocyturia after work shift predicted $\Delta eGFR_{cross-harvest}$ at eight out of nine occasions ($R^2$ ≥0.25) (figure 1).

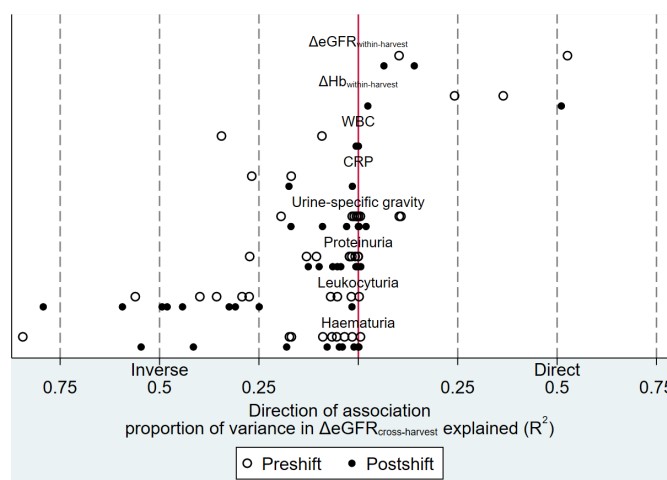

**Figure 1** Proportion of variance in $\Delta eGFR_{cross-harvest}$ explained by markers sampled during harvest. CRP, C reactive protein; eGFR, estimated glomerular filtration rate; Hb, haemoglobin; WBC, white blood cell. Each dot represents one sampling occasion. $\Delta eGFR_{within-harvest}$ and $\Delta Hb_{within-harvest}$ denote change in eGFR and Hb from baseline to measurement occasion during harvest. Exploratory dataset: Salvadorian male sugarcane workers, n=39.

Importantly, postshift leukocyturia was a much better predictor of $\Delta eGFR_{cross-harvest}$ than preshift (figure 1). Of the markers measured in blood or serum, the best was $\Delta Hb_{within-harvest}$, which had $R^2$ >0.24 at both preshift occasions and >0.5 at one of the postshift occasions (figure 1). USG, haematuria or proteinuria during harvest did not accurately predict $\Delta eGFR_{cross-harvest}$ (figure 1). No CPK levels were above 10 µkat/L and 95% were <6.44 µkat/L (upper reference interval 6.7 µkat/L).

Workers ever having pronounced leukocyturia (++) after a work shift during harvest had much worse $\Delta eGFR_{cross-harvest}$ (figure 2). Workers developing leukocyturia during harvest had lower preharvest eGFR and were slightly older (table 2).

Three out of four workers with ++ postshift leukocyturia during harvest had >20 g/L drop in Hb from baseline at one of the preshift measurements during harvest (online supplemental file 1), and those three workers had the largest $\Delta eGFR_{cross-harvest}$ declines. Workers, once developing postshift leukocyturia, often had such at several subsequent fortnightly measurement occasions (online supplemental file 1).

### Confirmation analysis

In the first and second harvests of the Adelante Study respectively, 196 out of 234 and 255 out of 301 participating male cutters finishing harvest work could be linked to routine dipstick measurements recorded by mill OH staff. These 451 unique harvest-worker observations occurred in 371 unique workers, as 80 workers were included in both harvests. The average number of postshift dipsticks recorded per worker was 4.3 and 2.8 during the first and second harvests, respectively. Of these workers, 73% did not have any postshift dipstick leukocyturia recorded,

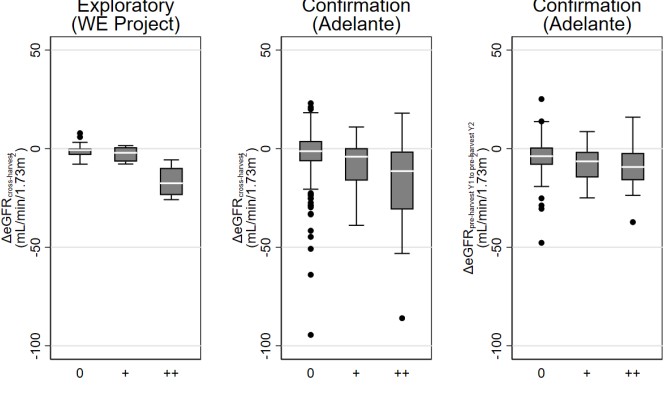

**Figure 2** Association between maximum postshift dipstick leukocyturia during harvest and $\Delta eGFR_{cross-harvest}$. eGFR, estimated glomerular filtration rate; WE, Worker Health and Efficiency. Exploratory dataset: El Salvador, n=39, all samples taken for research purposes. Confirmation dataset: Nicaragua, n=451 worker-harvests in 371 unique workers, serum samples taken for research purposes, urine samples from routine monitoring.

10% had at most + and 18% had ≥++ at some occasion (table 2). The median preharvest eGFR was lower in those recorded with leukocyturia during harvest, with a difference of 7 mL/min/1.73 m$^2$ between those with no recorded leukocyturia and those with ++ at some occasion (table 2), a difference increasing to 20 mL/min/1.73 m$^2$ at end-harvest. Workers recorded with leukocyturia during harvest were overall the same age as workers not recorded to have leukocyturia (table 2).

Maximum postshift leukocyturia was associated with $\Delta eGFR_{cross-harvest}$ (figure 2), with mixed linear regression estimating that for each increase in maximum postshift leukocyturia during harvest (ie, from 0 to +, and + to ++), $\Delta eGFR_{cross-harvest}$ was 6.5 mL/min/1.73 m$^2$ (95% CI 4.9, 8.1 mL/min/1.73 m$^2$) lower. The corresponding $\Delta eGFR_{cross-harvest}$ associated with each increase in maximum preshift leukocyturia category was –5.4 mL/min/1.73 m$^2$ (–3.9 to –6.9 mL/min/1.73 m$^2$).

The association between maximum postshift leukocyturia and cross-harvest eGFR declines was also seen when $\Delta eGFR$ was examined for reversibility between harvests

by calculating change from preharvest year 1 to preharvest year 2 (figure 2). Among 151 workers having both such measurements and who had a dipstick recorded during harvest 1, for each increase in maximum postshift leukocyturia during harvest, the $\Delta eGFR_{preharvest\ to\ preharvest}$ was lower by 2.4 mL/min/1.73 m$^2$ (95% CI 0.6, 4.2 mL/min/1.73 m$^2$). This indicates that leukocyturia during a harvest is associated with a larger net yearly eGFR loss, also after considering postharvest recovery of potentially reversible injury-associated eGFR loss during harvest.

As previously reported,[16 17] by interviewing workers who dropped out of harvest, thus missing end-harvest SCr sampling, follow-up success was achieved in all but 3% and 7% in years 1 and 2, respectively. This resulted in data for over 90% of the study group for the composite of measured and self-reported IKI$_{all}$ outcome. Among those with dipstick data recorded at any time during the harvest, postshift leukocyturia was strongly associated with IKI$_{all}$. IKI$_{all}$ occurred among 37/97 (38%) of those with ≥++ leukocyturia at any time during harvest compared with 33/383 (9%) of those who never had recorded leukocyturia.

## Association between leukocyturia and symptoms

Three-hundred forty-seven dipsticks among 49 workers were included. Workers with leukocyturia had more often experienced, and experienced more days of dark or little urine, cramps, headache, fever, dizziness or abdominal pain in the preceding 2-week period (table 3).

## DISCUSSION

Leukocyturia, measured using cheap and easy to analyse urine dipsticks, identified the workers with largest $\Delta eGFR_{cross-harvest}$ decline with relatively high accuracy in the exploratory dataset. Workers ever having leukocyturia during harvest had worse $\Delta eGFR_{cross-harvest}$ and $\Delta eGFR_{preharvest\ to\ preharvest}$ in the routine OH setting of the confirmatory dataset. These findings support the hypothesis that repeated/prolonged tubular inflammation likely is a key step in CKDnt progression.[4] Our results are consistent with previous reports of leukocyturia associated with SCr elevation in populations at risk of CKDnt.[3 7 8] Thus, the use of dipstick monitoring in workers at risk of CKDnt could be a valuable tool for OH services.

**Table 2** Age and eGFR at preharvest and end-harvest by maximum leukocyturia during harvest

| Dataset | | Exploratory | | | | Confirmation | | |
|---|---|---|---|---|---|---|---|---|
| Maximum leukocyturia during harvest | N | Age, median (IQR) years | eGFR, median (IQR) mL/min/1.73 m$^2$ | | N | Age, median (IQR) years | eGFR, median (IQR) mL/min/1.73 m$^2$ | |
| | | Preharvest | Preharvest | End-harvest | | Preharvest | Preharvest | End-harvest |
| – | 25 | 28 (24–35) | 122 (115–127) | 122 (113–126) | 328 | 28 (24–34) | 104 (86–117) | 101 (83–117) |
| + | 10 | 36.5 (26–57) | 107 (95–124) | 104 (96–121) | 43 | 28 (24–36) | 98 (82–110) | 91 (69–104) |
| ++ | 4 | 35 (31.5–46.5) | 103 (85–119) | 83 (61–109) | 80 | 27 (24–34) | 97 (79–116) | 81 (64–99) |

eGFR, estimated glomerular filtration rate.

**Table 3** Association between leukocyturia and symptoms in preceding 2 weeks

| | Leukocyturia | | | | Leukocyturia | | | |
| | 0 | + | ++ | | 0 | + | ++ | |
|---|---|---|---|---|---|---|---|---|
| Total samples (N) | 303 | 32 | 12 | | 303 | 32 | 12 | |
| | | | | OR* per symptomatic day (for increase in postshift leukocyturia, per symptomatic day in 2 preceding weeks (95% CI)) | | | | OR* for ever symptomatic (for increase in postshift leukocyturia, presence of symptom in 2 preceding weeks (95% CI)) |
| Symptom | Number of days with symptom in preceding 2-week period Mean (SD) | | | | At any day in preceding 2-week period experiencing symptom N (%) | | | |
| Extremely dry mouth | 1.2 (2.4) | 2.1 (3.7) | 4.8 (6.7) | 1.1 (1.0 to 1.3) | 104 (34%) | 12 (38%) | 5 (42%) | 1.1 (0.4 to 3.2) |
| Dysuria | 1.1 (2) | 1.8 (2.5) | 0.7 (1.6) | 1.1 (0.9 to 1.3) | 128 (42%) | 19 (59%) | 2 (17%) | 1.3 (0.5 to 3.2) |
| Little urine | 0.6 (1.8) | 1.7 (2.9) | 2.2 (5.1) | 1.3 (1.1 to 1.5) | 63 (21%) | 15 (47%) | 2 (17%) | 3.9 (1.4 to 10.4) |
| Dark urine | 1.2 (3) | 1.9 (3.2) | 1.5 (3.5) | 1.1 (1.0 to 1.3) | 70 (23%) | 15 (47%) | 3 (25%) | 5.3 (2.0 to 14.3) |
| Cramps | 0.6 (1.7) | 1.3 (2.9) | 0.9 (2.9) | 1.3 (1.1 to 1.5) | 55 (18%) | 12 (38%) | 2 (17%) | 5.8 (1.8 to 18.3) |
| Headache | 1.3 (2.2) | 2.8 (4.1) | 1.6 (1.5) | 1.3 (1.1 to 1.5) | 134 (44%) | 22 (69%) | 9 (75%) | 5.2 (1.8 to 14.5) |
| Fever | 0.3 (0.8) | 1.6 (3.2) | 3.2 (5.4) | 1.5 (1.2 to 1.9) | 41 (14%) | 14 (44%) | 5 (42%) | 3.8 (1.5 to 9.3) |
| Nosebleed | 0 (0.3) | 0.2 (0.9) | 0 (0) | 2.2 (0.9 to 5.4) | 8 (3%) | 2 (6%) | (%) | 2.8 (0.3 to 29.5) |
| Tremor | 0.2 (0.8) | 0.8 (2.9) | 0.3 (0.6) | 1.4 (1.1 to 1.9) | 31 (10%) | 4 (13%) | 2 (17%) | 1.8 (0.5 to 6.3) |
| Hand/Foot inflammation | 0.1 (1) | 0.1 (0.5) | 0.8 (1.6) | 1.1 (0.8 to 1.5) | 13 (4%) | 1 (3%) | 3 (25%) | 2.8 (0.5 to 14.8) |
| Abdominal pain | 0.3 (0.9) | 0.6 (1) | 1.3 (2.7) | 1.7 (1.2 to 2.4) | 42 (14%) | 11 (34%) | 3 (25%) | 3.2 (1.1 to 9.1) |
| Feel like vomiting | 0.3 (0.9) | 0.4 (0.7) | 0.1 (0.3) | 1.4 (0.8 to 2.4) | 47 (16%) | 9 (28%) | 1 (8%) | 2.9 (0.9 to 8.8) |
| Vomiting | 0.1 (0.4) | 0.1 (0.3) | 0.1 (0.3) | 2.1 (0.7 to 6.6) | 17 (6%) | 3 (9%) | 1 (8%) | 4.8 (0.9 to 24.9) |
| Diarrhoea | 0.1 (0.4) | 0.1 (0.4) | 0 (0) | 0.7 (0.2 to 2.5) | 11 (4%) | 2 (6%) | (%) | 0.4 (0.0 to 3.1) |
| Breathing difficulties | 0.4 (1.4) | 0.1 (0.5) | 0 (0) | 0.7 (0.3 to 1.5) | 43 (14%) | 2 (6%) | (%) | 0.4 (0.1 to 2.1) |
| Wheezing | 0.5 (1.7) | 0.2 (0.8) | 0.5 (1.4) | 1.0 (0.7 to 1.3) | 36 (12%) | 2 (6%) | 2 (17%) | 0.9 (0.2 to 3.5) |
| Fainting | 0 (0.1) | 0 (0) | 0 (0) | – | 2 (1%) | (%) | (%) | – |
| Palpitations | 0.4 (1.3) | 0.5 (1.6) | 0.4 (1.4) | 1.0 (0.8 to 1.3) | 38 (13%) | 4 (13%) | 1 (8%) | 0.6 (0.2 to 2.3) |
| Weakness | 0.8 (2) | 1.2 (1.9) | 3.7 (5.7) | 1.1 (1.0 to 1.3) | 72 (24%) | 10 (31%) | 5 (42%) | 1.6 (0.6 to 4.1) |
| Disorientation | 0.2 (1) | 0.1 (0.4) | 0.3 (0.6) | 1.0 (0.6 to 1.6) | 18 (6%) | 1 (3%) | 2 (17%) | 2.4 (0.4 to 14.1) |
| Dizziness | 0.2 (0.9) | 0.5 (1.1) | 2 (4.7) | 1.5 (1.0 to 2.1) | 32 (11%) | 8 (25%) | 2 (17%) | 4.1 (1.3 to 12.7) |

*OR from multinomial mixed effects logistic regression model.

Postshift leukocyturia was the marker which most consistently explained a large proportion of variance in $\Delta eGFR_{cross-harvest}$, with a clearly worse $\Delta eGFR_{cross-harvest}$ for those with leukocyturia ≥++. However, the predictive performance of $\Delta Hb_{within-harvest}$ was comparable, and no formal comparison have been performed here due to the relatively small study size.

Leukocyturia was associated with experiencing several symptoms, both indicative of potentially causative factors[4] such as dehydration (low urine output and dark urine), systemic inflammation/heat stress (feeling hot/fever), intense muscle work/injury (cramps) and more general symptoms (headache, dizziness and abdominal pain). This is consistent with previous findings of fever being associated with kidney injury among sugarcane workers.[3 4] Testing heat-stressed workers, when presenting with similar symptoms, for leukocyturia may identify kidney injury at an early and possibly reversible stage.

Although dark urine was associated with leukocyturia, CPK levels that might serve as indicators of rhabdomyolysis, were essentially normal.[4] Thus, frank rhabdomyolysis does not seem to be a main cause of dark urine or kidney injury in this population.

Change in blood Hb from baseline, which can be measured using point-of-care (POC) methods at a considerably lower cost than SCr, was about as useful

as $\Delta eGFR_{within-harvest}$ for predicting $\Delta eGFR_{cross-harvest}$. This is consistent with the possibility that Hb is an early indicator of reduced erythropoietin expression following transformation of kidney fibroblasts to profibrotic myofibroblasts[10–12] and anaemia as a risk factor for progression from AKI to CKDnt.[9] Hb should be evaluated as a source of valuable information on risk of disease progression among workers with leukocyturia.

USG was not useful for predicting $\Delta eGFR_{cross-harvest}$. USG >1.018 has been proposed to identify heat-stressed workers at risk of kidney injury, based on a higher concentration of the kidney injury biomarker nephrin in urine in these workers.[18] While we agree that proper hydration is important for preventing kidney injury among heat-stressed workers, and hydration was a key component of the WE and Adelante projects, it is obvious that concentrations of kidney injury proteins in urine will increase as the urine becomes more concentrated overall (ie, USG increase). Furthermore, USG spot measurements reflect very short-term (hours) hydration practices and, in addition, low USG levels may be caused by decreased urine concentration ability secondary to tubular injury.

## LIMITATIONS

Before implementing our findings into a workforce kidney injury monitoring programme, several questions which are not answered in the present study need to be addressed. The small size and observational character of the study limit our possibilities to provide strong evidence on which biomarkers to monitor, appropriate cut-off values and the effect of actions taken in response to these markers. As previously mentioned, monitoring the workforce for kidney injury using mid-harvest SCr is already implemented at Ingenio San Antonio where our confirmation dataset was collected. Workers with kidney injury are sent for medical care or long-term rest, or assigned to a less intense daily quota or different job group. This may lead to a bias as workers with kidney injury during harvest have a higher risk of missing the final harvest evaluation or have had their workload reduced. The likely direction of bias introduced by ongoing screening at Ingenio San Antonio is towards the null, decreasing the true association between leukocyturia during harvest and $\Delta eGFR_{cross-harvest}$, as this is the common direction of bias resulting from healthy worker selection.[19–21]

Ideally, a cost-benefit analysis should be conducted before implementation of a leukocyturia or Hb surveillance programme, comparing the costs with health benefits of early detection in light of the predictive accuracy seen here. However, the low-cost technology, with one urine dipstick typically costing ~US$0.30, and an automated dipstick reader a few hundred USDs, are promising. The impact of early detection and exposure cessation however needs to be studied directly before confidently recommending broad application of leukocyturia surveillance.

These results apply to the interpretation of leukocyturia and anaemia in male, heat-stressed workers at very high risk of CKDnt with a high pretest probability of kidney injury. The same indicators in other settings would more likely be indicators of other aetiologies. Specifically, a higher probability of UTIs and iron-deficiency anaemia among women than men means leukocyturia and anaemia likely is less specific of kidney injury among heat-stressed women. However, we cannot evaluate this in the present study due to the small number of women working as cutters, and especially the small number having large $\Delta eGFR_{cross-harvest}$ declines.[16 17] The two cohorts used here are not directly comparable as there were different prehiring and leukocyturia screening procedures. The predictive ability of the biomarkers were not adjusted for, for example, age or baseline eGFR. Workers developing leukocyturia had slightly lower eGFR than those not developing leukocyturia, meaning that practitioners may need to monitor these workers more carefully. There may also be other factors predisposing heat-stressed workers to leukocyturia and/or kidney function decline that warrant increased awareness, such as diabetes and hypertension, but these were not addressed in the present study and are uncommon in these populations.[15 17]

Although limiting further heat strain from job effort seems warranted in workers with leukocyturia and declining Hb or eGFR, the beneficial effects on long-term kidney outcomes of sick leave, ending employment or transferring workers to a job associated with less heat strain is at present unclear. As these interventions may also result in large economic difficulties for workers where there is little or no social security network or alternative employment, the potential effectiveness of such screening procedures should be evaluated in research studies that include providing due compensation or alternative employment to workers suspected of having early kidney injury. Employers should not aim for individual-level detection of early stages of kidney injury as their primary way of protecting workers from CKDnt. Ensuring adequate working practices preventing the entire workforce also from early stages of kidney injury should remain a priority. Therefore, monitoring of early injury markers may ideally be a tool for evaluating primary prevention practices.

This study does not address what causes leukocyturia. No urine cultures were performed to formally rule out UTI. No worker in El Salvador and only 1.5% of Nicaraguan postshift samples with leukocyturia ≥++ had positive nitrite tests. This implies that regular UTIs seem very unlikely considering that most studies find that nitrite has approximately 50% sensitivity for UTI.[22 23] Furthermore, in two previous studies among workers at the same Nicaraguan mill, 0 of 114 and 0 of 70 urine cultures among male sugarcane workers were positive, although 30 and 29, respectively, of these had leukocyturia.[8 24] Dysuria (known locally as *chistata*) has been considered a potential indicator of kidney injury from micro-crystalluria in some previous studies.[8 25 26] In this study, worker reports of dysuria were not associated with leukocyturia. Whether heat stress causes leukocyturia,

and whether leukocyturia is associated with a rise in kidney-specific injury biomarkers, deserve further studies.

Creatinine, Hb, WBC and CRP were not assessed using POC devices but in hospital laboratories. We expect somewhat poorer performance using POC devices. A research study among Guatemalan sugarcane workers have validated the use of POC devices for creatinine in the field settings, finding that an adjustment factor needs to be applied to correct results.[27]

Although studies indicate that relatively acute events of decreases in eGFR (across a harvest period) are likely a driver of CKDnt,[9 17 28] the long-term relevance of cross-harvest eGFR declines is still under study. However, we have also shown that the eGFR trajectory from the baseline 1 year to the baseline the next (follow-up 6 months *after* end of harvest) is worse among those with leukocyturia, suggesting that the 5–6 months documented eGFR decline is not immediately reversible. We have also previously reported that tubuli-specific kidney injury biomarkers such as Kidney Injury Molecule-1 (KIM-1) and Monocyte Chemoattractant Protein-1 (MCP-1) increase as eGFR decline increases among sugarcane workers during harvest, indicating that this condition, and probably also the associated leukocyturia, is indicative of kidney injury.[12]

## CONCLUSION

Markers which allow early identification of kidney injury at the individual level could contribute importantly towards prevention by risk factor elimination as well as learning about the causes of and halting the CKDnt epidemic. The potential utility of several candidate markers that are available using POC analyses was explored in this study, with postshift leukocyturia identified as the most promising candidate. However, even if early detection of CKDnt risk in individual workers may become an effective tool for prevention, the need to understand and act against the underlying pathophysiological mechanisms and unhealthy work practices driving the CKDnt epidemic remains.

**Author affiliations**
[1]La Isla Network, Washington, District of Columbia, USA
[2]Occupational and Environmental Medicine, School of Public Health and Community Medicine, Institute of Medicine, Sahlgrenska Academy, University of Gothenburg, Göteborg, Sweden
[3]Institute of Environmental Medicine, Unit of Occupational Medicine, Karolinska Institute, Stockholm, Sweden
[4]University of Massachusetts Lowell, Lowell, Massachusetts, USA
[5]Department of Laboratory Medicine, Lund University, Lund, Sweden
[6]Occupational Health Management, Ingenio San Antonio/Nicaragua Sugar Estates Limited, Chichigalpa, Nicaragua
[7]Occupational and Environmental Medicine, Sahlgrenska University Hospital, Gothenburg, Sweden

**Acknowledgements** We are grateful to the sugarcane workers who provided biological samples and answered questionnaires, and the field research staff and health workers who collected these. Ilana Weiss coordinated field research activities both in the Salvadoran and Nicaraguan cohorts and Theo Bodin had a leading role in collecting data and samples from the Salvadoran cohort.

**Contributors** DW, CW, JG, KJ, UE and EH planned the study. DC, UE, JG and CW participated in data collection. EH was responsible for statistical analyses and drafting the manuscript. DW was the principal investigator for the main research project, and KJ supervised EH in this specific substudy. All authors contributed to interpretation of data and critically revised the manuscript for important intellectual content. EH is the guarantor.

**Funding** The data were collected through the intervention studies: Worker Health and Efficiency Program Working Group (WE), El Salvador, and Adelante Initiative, Nicaragua. The WE study was supported by a grant from the Dutch National Postcode Lottery to Solidaridad, while the El Ángel sugar mill provided funds to pay for the intervention components studied. The Adelante study was funded by the Stavros Niarchos Foundation, and the German Investment Corporation (DEG) and German Ministry for Economic Development and Cooperation (BMZ), and the Ingenio San Antonio (ISA) sugar mill via the DEG and BMZ's DeveloPPP. de Programme. The intervention parts of these studies (eg, shade tents, water containers) were provided by the sugarcane mills where the studies were carried out. Neither company nor other sponsors had any role in the design, execution, interpretation or writing of the study except for the input from DC who is Occupational Safety and Health physician at ISA and provided access to monitoring data and information and perspectives on current screening and individualised prevention practices. Data analysis and scientific writing was funded within a Belmont Forum grant from the Swedish Research Council FORTE (dnr 2019-01548, acronym PREP) and in-kind funding from University of Gothenburg, Sweden and Lund University, Sweden.

**Competing interests** EH, CW, DW, JG, KJ and UE declare no conflicts of interest. DC is employed by ISA, but the views expressed here are not his and not the views of his employer.

**Patient and public involvement** Patients and/or the public were not involved in the design, or conduct, or reporting, or dissemination plans of this research.

**Patient consent for publication** Not applicable.

**Ethics approval** This study was approved by the Comité de Ética para Investigaciones Biomédicas (CEIB), Facultad de Ciencias Médicas, Universidad Nacional Autónoma de Nicaragua (UNAN—León) (FWA00004523/IRB00003342) and the National Ethics Committee for Clinical Research (Comité Nacional de Ética de Investigación de Salud) of El Salvador (OHRP IRB No. 0005660, FWA No. 00010986). The biochemical investigations carried out at the Division of Clinical Chemistry and Pharmacology at Lund University in Sweden were approved by the Regional Ethical Review Board in Lund (reg no 2018-256 and 2016-60). Participating workers provided written informed consent.

**Provenance and peer review** Not commissioned; externally peer reviewed.

**Data availability statement** Data are available on reasonable request. Data are available from the corresponding author on reasonable request.

**ORCID iD**
Erik Hansson http://orcid.org/0000-0002-9779-6820

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
