## [Reviewer comments · BMJ Open]

ARTICLE DETAILS

TITLE (PROVISIONAL)	Point-of-care biomarkers for prediction of kidney function trajectory among sugarcane cutters - a comparative test accuracy study
AUTHORS	Hansson, Erik; Wesseling, Catharina; Wegman, David; Ekström, Ulf; Chavarria, Denis; Glaser, Jason; Jakobsson, Kristina

VERSION 1 – REVIEW

REVIEWER	Martínez-Castelao, Alberto
REVIEW RETURNED	31-Mar-2022

GENERAL COMMENTS	The paper exposes the importance of leukocyturia as a maker for kidney damage and CKD in two populations of sugar cane cutters workers.. My MAJOR CONCERN is that the authors did no clearly specified if some workers, especially those having dark urine, could suffer from rhabdomyoysis. I can imagine that no CPK determinations were done. In my opnnion a brief comment on taht aspect may be do in the discussion. Other MINOR comments are added in blue or yellow colours on the original pdf.
---

REVIEWER	Coca, Steven Mount Sinai Hospital, Medicine Nephrology
REVIEW RETURNED	19-Apr-2022

GENERAL COMMENTS	This study examines some simple clinical markers to predict eGFR decline in sugarcane workers in El Salvador and Nicaragua. There seemed to be an association of leukocyturia with eGFR decline. The discussion points out several limitations. however, I am concerned about the following issues: 1. Lots of drop-out and loss to follow-up, especially in the confirmation Nicaraguan cohort2. While a linear mixed model for a 3 level exposure variable (0, +, 2+) of leukocyturia?3. No multivariable adjustment performed; all univariable analyses4. No full predictive model employed for eGFR decline5. No assessment of the shade tents and hydration on changes in leukocyturia6. No assessment of kidney injury biomarkers7. No assessment of the reversibility of the eGFR decline
---

VERSION 1 – AUTHOR RESPONSE

Reviewer 1 Major comment 1/1: My MAJOR CONCERN is that the authors did not clearly specify if some workers, especially those having dark urine, could suffer from rhabdomyolysis. I can imagine that no CPK determinations were done. In my opinion a brief comment on that aspect may be done in the discussion.

We have CPK data. CPK levels before and after multiple work shifts in the El Salvador study group has previously been reported in Wegman et al 2018, SJWEH. We now more clearly cite this on pages 5-6 lines 110-113, specifically mentioning CPK.

CPK was not in the focus of this publication as it is rarely available for point-of-care analysis. As previous studies (reviewed for Mesoamerican sugarcane cutters and analyzed for the Nicaraguan population in a publication by us (<https://doi.org/10.3390/nu12061639>)) have not found a clear association between elevated CPK and kidney injury at an individual level in populations at risk of CKD we did not consider this further here. For your interest, CPK concentrations in this material have no relation with kidney function loss, dark urine or leukocyturia. We have added a sentence reporting on CPK levels in the Salvadoran group (page 10, lines 218-219) and a brief paragraph interpreting this (page 16, lines 303-305).

The other comments by this reviewer are minor (mostly about explaining abbreviations) and have been addressed in the revised version. However, we did expand the explanation on why we excluded the one female worker in the Salvadoran study (pages 6, lines 115-118) as well as the small number of female workers in the Nicaraguan study (page 8, lines 161-163).

Reviewer 2.

1. Lots of drop-out and loss to follow-up, especially in the confirmation Nicaraguan cohort

In the Nicaraguan cohort we followed workers who did not attend end-harvest testing by home visits, without collecting biological samples as this was not logistically feasible. Many of these workers had dropped out due to kidney injury.

We have simplified our reporting on these self-reported kidney injury cases (many confirmed by or detected by the mill, see Hansson et al 2019 and Glaser et al 2020, as well as Fischer et al 2017) by revising the text sections on page 9 lines 184-189 and page 13, lines 267-273. In fact, the proportion of workers who dropped out completely during the Nicaraguan study was very low, at 3% and 7% in each of the years, but we do not have biological data for all. This is acknowledged in the discussion. However, as presented in the discussion, one must also consider the likely direction of bias resulting from this loss of follow-up. Here we would predict the direction would most likely *reduce* any observed association of kidney injury with leukocyturia as symptomatic workers with leukocyturia and a more severe type of kidney injury are more likely to drop out of the study, in line with the bias resulting from healthy worker selection effects in general. (Eisen 1995, Eisen, Picciotto et al. 2001, Buckley, Keil et al. 2015)

Buckley, J. P., A. P. Keil, L. J. McGrath and J. K. Edwards (2015). "Evolving methods for inference in the presence of healthy worker survivor bias." *Epidemiology* **26**(2): 204-212.

Eisen, E. A. (1995). "Healthy worker effect in morbidity studies." *Med Lav* **86**(2): 125-138.

Eisen, E. A., S. Picciotto and J. M. Robins (2001). Healthy Worker Effect. *Encyclopedia of Environmetrics*.

2. While a linear mixed model for a 3 level exposure variable (0, +, 2+) of leukocyturia?

We suppose that the reviewer means "Why" rather than "while".

"Linear" in "linear mixed model" refers to the outcome variable being continuous, as an alternative to a binomial or Poisson model, where the outcome would be a binary variable or a count. The type of exposure variable, which the reviewer is referring to and implying that "linear" mixed model is

incorrect, is not relevant in linear mixed models (the exposure variable can be continuous, binary or categorical).

If supposing the reviewer's question here is why is leukocyturia parameterized as a linear variable when it is an ordinal variable? This is a reasonable question. One advantage of treating an ordinal explanatory variable as linear is that statistical power is increased as equal intervals are assumed and the number of parameters is reduced. The simple alternative, treating each value as a category with no inherent relationship to other values can also easily be performed in order to provide a conservative estimate which has less power than available as it ignores existing information on the order of ordinal variable values.

These are the results of such a model treating leukocyturia as a categorical variable:

Max post-shift leukocyturia	Cross-harvest eGFR regression coefficient
0	Reference
+	-5.6 (-1.4, -10.0)
++	-13.3 (-10.0,-16.6)

i.e. the steps between each increase in leukocyturia are quite similar, making it reasonable to use the linear form.

Parameterizing ordinal variables as linear is common but has been criticized by contemporary Bayesian statisticians. For example these statisticians have recently developed R packages (<https://bpspsychub.onlinelibrary.wiley.com/doi/abs/10.1111/bmsp.12195>), making it possible to use monotonic effects (assuming each change in ordinal category is in the same direction, but allowing the size of each step to vary) regression for ordinal predictors. While this may have been possible, we did not use this as we did not want to introduce unnecessary complexity which would detract from the study purpose. Furthermore, it was more transparent to use the same approach for all possible predictor variables for the Salvadoran exploratory data. Further, we believe most clinicians would interpret the leukocyturia ordinal variable as a gradient and we wanted to provide an analysis which would more transparently indicate if that gradient was something to be concerned about.

That said, we did evaluate the difference in interpretation of output from this novel statistical approach (monotonic effects mixed linear model) that we have not reported and the standard (mixed linear model) that we do report in the manuscript for the Nicaraguan dataset. In the monotonic effects linear mixed model, each increase in maximum post-shift leukocyturia (from 0 to + and + to ++) was associated with an average -6.1 ml/min/1.73 rather than a -5.4 ml/min/1.73 eGFR as found in the linear mixed model treating leukocyturia as a linear variable.

To summarize, if the reviewer is asking why we parameterized leukocyturia as a linear variable when it is ordinal, the reason is that this is an often used approach we believe would be more easily understandable in line with how clinicians would interpret the variable. It also yields similar results as approaches which are more novel and complicated or more conservative approach.

3. No multivariable adjustment performed; all univariable analyses

While true, the reviewer does not indicate why a multivariable adjustment would be desired. The purpose of the study was to identify useful predictors in an applied setting, not to identify causal relationships between some factors while adjusting for others. It is for this reason that we specifically did not include in the aims paragraph that identifying causal relationships was an objective of this manuscript. Further, the discussion was not written in terms of a causal explanation for the findings. Furthermore, the study population is very homogenous in terms of commonly considered confounders. All were men, the great majority between 25-45 years with the same occupational exposure living with very similar socioeconomic exposures in the same geographical setting. As the reviewer does not indicate any confounders that were likely but not accounted for in order to address the stated study objectives we can only offer the general reasons for not considering multivariable adjustments to address our objectives.

4. No full predictive model employed for eGFR decline

It is not evident what the reviewer means by a "full predictive" model as the aim of the paper is definitely not "full prediction". We suggest that a "full predictive" model is not realistic so imagine the reviewer had some unstated concern related to the issue raised in point 3.

It can be debated whether the degree of prediction is sufficient in relation to the cost of conducting the screening proposed, and we have added this as a limitation of our work in the discussion section, on page 17 lines 334-339.

5. No assessment of the shade tents and hydration on changes in leukocyturia

This was not the aim of this manuscript. Intervention effects on eGFR and a creatinine-based definition of kidney injury have been reported in previously published papers referenced in the manuscript (Glaser 2020, Wegman 2018). We do not wish to add this analysis to the present manuscript as it is not related to the aim of the paper and likely would detract from that purpose. Incidence of leukocyturia was clearly reduced in the second Nicaragua year when a more comprehensive rest-shade-hydration intervention was implemented, in line with the association between leukocyturia and eGFR decline.

6. No assessment of kidney injury biomarkers

While not relevant to the aim of this manuscript the assessment of kidney injury biomarkers in the cohort has been published separately (Hansson 2021: <https://oem.bmj.com/content/early/2022/02/13/oemed-2021-107989>). There we have described a very strong association between kidney function decline across harvest and elevation in KIM-1 and MCP-1 in the same time period in the Nicaraguan cohort. This information, with a citation to Hansson 2021 is now included in the resubmitted version (page 19, lines 393-396). Performing further such analysis within Salvadoran samples would be interesting (as noted in the discussion, page 18, line 367-368) but is not possible at the moment and is not related to the stated objective of the study reported in the manuscript.

7. No assessment of the reversibility of the eGFR decline

We thank the reviewer for calling attention to this important point and have included this as a limitation (page 19, lines 389-391)

We currently have follow-up data from the Nicaraguan cohort members who worked both year 1 and year 2 (year 1 end-harvest to year 2 pre-harvest) that enables addressing this question, and we include an analysis of this in the revised version as reported on page 12, lines 259-266 and page 18-19 lines 373-380, as well as in a revised figure 2. Having leukocyturia during harvest year 1 was associated with a larger decrease from pre-harvest year 1 to pre-harvest year 2, implying that leukocyturia was associated with worse eGFR which was not reversed by 6 months after harvest work ended. Whether further restitution of kidney function occurs after this we cannot determine at this stage. We are aiming to follow this cohort but can at present do not have results in hand to undertake any such analyses.

VERSION 2 – REVIEW

REVIEWER	Martínez-Castelao, Alberto
REVIEW RETURNED	05-Sep-2022
GENERAL COMMENTS	The changes introduced by the authors with regards to the previous version are adequate.